# Discontinuous Hydration Cycles with Elicitors Improve Germination, Growth, Osmoprotectant, and Salt Stress Tolerance in *Zea mays* L.

Kleane Targino Oliveira Pereira [1], Salvador Barros Torres [1], Emanoela Pereira de Paiva [1], Tatianne Raianne Costa Alves [1], Maria Lilia de Souza Neta [1], Jefferson Bittencourt Venâncio [1], Lauter Silva Souto [2], Clarisse Pereira Benedito [1], Tayd Dayvison Custódio Peixoto [1], Miguel Ferreira Neto [1], Nildo da Silva Dias [1] and Francisco Vanies da Silva Sá [1,*]

[1] Department of Agronomic and Forest Science, Federal Rural University of the Semi-Arid—UFERSA, Mossoró 59625-900, Brazil
[2] Center for Agro-Food Science and Technology, Federal University of Campina Grande, Pombal 58840-000, Brazil
* Correspondence: vanies_agronomia@hotmail.com; Tel.: +55-(83)9-9861-9267

**Abstract:** Saline stress impairs germination and initial plant growth. However, discontinuous hydration cycles induce osmotic tolerance in seeds and can improve the response of maize seeds to saline stress. The objective of this study was to evaluate the action of discontinuous hydration cycles with different salt stress tolerance elicitors on germination, growth, and osmotic adjustment of maize cultivars. Maize seeds of BR 206 and BRS 5037 Cruzeta cultivars were subjected to the following treatments: 0.0 mmol of NaCl (control), 250 mmol of NaCl (salt stress), salt stress + three discontinuous hydration cycles (DHCs) of seeds in water, salt stress + DHCs with gibberellic acid, salt stress + DHCs with hydrogen peroxide, salt stress + DHCs with salicylic acid, and salt stress + DHCs with ascorbic acid. Salt stress reduced the germination, growth, and biomass accumulation in maize seedlings—the BR 206 cultivar outperformed BRS 5037 Cruzeta. Discontinuous hydration cycles with water failed to improve the salt stress tolerance of maize seeds. However, discontinuous hydration cycles with gibberellic acid, hydrogen peroxide, and salicylic acid promoted salt stress tolerance in maize due to increased synthesis of osmoprotectants. Our results revealed salicylic acid is appropriate for discontinuous hydration cycles in maize seeds.

**Keywords:** *Zea mays* L.; salicylic acid; gibberellic acid; $H_2O_2$; salinity

## 1. Introduction

*Zea mays* L. is one of the world's most commercially relevant annual crops. Between 2000–2020, Brazil maintained its position as the third largest producer of maize, behind the United States and China. In 2020, production was 100 million tons [1]. The expected 2020/21 season production is 112.3 million tons [2]. Some factors may influence this species' yield, especially in semi-arid regions. Water deficit, associated with rainfall irregularities and dry climate, followed by salinity, are the main threats to plant growth and agricultural yield [3,4].

Semi-arid regions commonly have saline and sodic soils, which affect crops' germination process and growth as they restrict water absorption and contain toxic Na$^+$ and Cl$^-$ ions [5,6]. High concentrations of these ions in the tissues hinder the mobilization of nutrient reserves, preventing germination and embryo growth. To acclimate to salt stress, plants adjust ionically by compartmentalizing organic ions in vacuoles or excluding ions in the roots [7,8].

Osmotic adjustment occurs through the accumulation of organic solutes, such as L-proline, soluble amino-N, soluble sugars, and other osmoprotective agents, in the cytosol to

reduce the cellular osmotic potential [6,9]. Ionic toxicity and osmotic stress are direct effects that can cause oxidative stress and several secondary stresses. Oxidative stress causes changes in plant metabolism, leading to excessive production of reactive oxygen species (ROS) that cause damage to cytoplasmic membranes and even cell death [6,10].

Plants produce osmoprotective agents to balance metabolic changes and reduce the effects caused by salts; among them, sugars and amino acids are produced in more significant quantities and accumulated in plant cells. *Proline* is an osmoprotectant that acts on antioxidant activity through the accumulation in chloroplasts of plant cells in response to the effects of environmental stresses, including salt stress. Thus, this amino acid performs several functions, such as stabilizing cellular structures and signaling the production of enzymes to eliminate ROS, reducing the deleterious effects of abiotic stresses [11,12].

Recent studies have shown that using hydropriming, organic acids, and hydrogen peroxide in pre-germination treatment in seeds attenuates the effects of salt stress [13–15]. Discontinuous hydration cycles (DHCs), known as water memory, constitute a pre-germination technique adopted to mitigate abiotic stresses [16–18]. DHCs are mainly studied in forest species to minimize water stress, but recently some studies have been conducted with agricultural species, such as Sorghum [*Sorghum bicolor* (L.) Moench.] [19] and fruit crops such as *Annona squamosa* L. [20]. These studies demonstrate that DHCs improve the tolerance of seeds and seedlings to dehydration.

The discontinuous hydration process occurs naturally in arid and semi-arid regions. In these regions, when water becomes available (rainy season), the seeds begin the imbibition process, which is quickly interrupted when the water becomes unavailable (dry spells). Seeds start losing water to the environment slowly, without causing damage to internal tissues. This process can occur in cycles until hydration is sufficient to initiate the metabolic activities of the seeds and consequently continue the germination process [16–18]. Thus, the species acclimate to climatic adversities and respond efficiently to abiotic stresses that occur in the field.

We hypothesize that discontinuous hydration cycles with stress tolerance elicitors can mitigate salt stress in maize seeds. We also hypothesized that discontinuous hydration cycles in drought-tolerant maize seeds could induce salt stress tolerance. Maize tolerance to salinity is influenced by the concentration of salts, time of exposure to stress, phenological stages, and genotype [13,21]. Therefore, the mechanisms of tolerance and changes in metabolism, seed germination, and seedling growth are expressed differently depending on genotype and environment [22]. Thus, this study aimed to evaluate the action of discontinuous hydration cycles with different salt stress tolerance elicitors on germination, growth, and osmotic adjustment of maize cultivars.

## 2. Materials and Methods

### 2.1. Location and Acquisition of Seeds

The experiment was conducted between June and December 2019 in the Seed Analysis Laboratory and Plant Physiology Laboratory of the Federal Rural University of the Semi-Arid Region (UFERSA), in the municipality of Mossoró/RN, Brazil (5°11′ S and 37°20′ W, and 18 m altitude).

Maize seeds, of cultivars BR 206 and BRS 5037 Cruzeta, were obtained from the private company GranSafra Sementes and EMPARN (Empresa de Pesquisa Agropecuária do Rio Grande do Norte), respectively. After receipt, they were stored in a controlled environment (16–18 °C and 40% relative humidity) throughout the experimental phase. We chose these maize cultivars for their characteristics favorable to grain production, biomass, and drought tolerance. These cultivars are indicated for production in the Brazilian semi-arid region, and in addition to being drought-stress tolerant, they can be salt-stress tolerant. Cultivar BR 206 is a drought-tolerant double hybrid, with an aptitude for grain yield, with an average yield of 8800 kg ha$^{-1}$. The BRS 5037 Cruzeta cultivar is drought tolerant and has an aptitude for biomass and grain production, with an average grain yield of 4290 kg ha$^{-1}$.

### 2.2. Imbibition Curve and Experimental Design

Initially, the moisture content of the seeds was quantified using the oven method at $105 \pm 3$ °C for 24 h [23], using two replicates of $4.5 \pm 0.5$ g. The moisture content was calculated based on the wet mass and expressed as a percentage.

The imbibition curve was obtained with two replicates of 50 seeds. These were weighed on a digital analytical scale (0.001 g) before imbibition and after each predetermined time interval until the emergence of the primary root. Imbibition was performed via immersion in water, with the seeds arranged in a beaker with 100 mL of distilled water and kept in germination chambers at 25 °C. Initially, seed weighing was performed every hour for eight hours of imbibition. Then, weighing was performed every two hours until thirty-four hours of imbibition. Finally, seed weighings were performed every four hours until fifty-two hours of hydration, when primary root protrusion was observed in 50% of the seeds of each replicate (Figure 1).

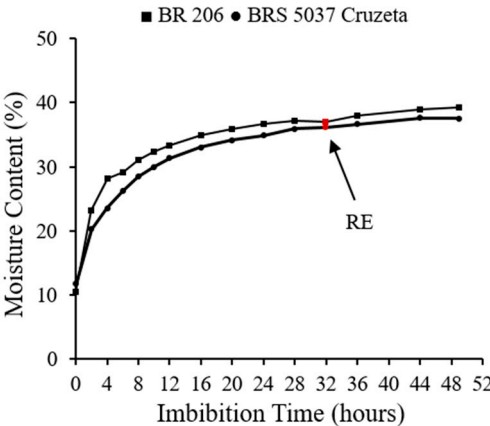

**Figure 1.** Imbibition curve of maize (*Zea mays* L.) seeds, cultivars BRS 5037 Cruzeta and BR 206, by immersion in water. RE = root emergence.

The experiment was conducted in a completely randomized design, following a $2 \times 7$ factorial arrangement, with four replicates of 50 seeds. We used two maize cultivars (BR 206 and BRS 5037 Cruzeta) plus seven saline stress combinations with three discontinuous hydration cycles (DHCs) of seeds with stress elicitors. The seven combinations were: 1—0.0 mmol of NaCl (control); 2—250 mmol of NaCl (salt stress); 3—salt stress + DHCs in water; 4—salt stress + DHCs with gibberellic acid (50 μM $GA_3$) [24]; 5—salt stress + DHCs with hydrogen peroxide (5 mmol ($H_2O_2$); 6—salt stress + DHCs with salicylic acid (50 μM SA); and 7—salt stress + DHCs with ascorbic acid (50 μM ASC) [24].

Maize seeds underwent three hydration–dehydration cycles. In the hydration process, the 200 seeds, comprising the four replications of 50 seeds, were soaked in 180 mL of the elicitor agent for two hours in a germination chamber at 25 °C in the dark. Subsequently, the seeds underwent twelve hours of dehydration at room temperature (28–30 °C) [16–18]. The DHCs were defined according to the data obtained in the imbibition curve (Figure 1), and the dehydration time was based on preliminary tests. At the end of the DHCs, the seeds had a moisture content of around 25%.

### 2.3. Germination and Seedling Length

After the DHCs, the seeds were sown in a paper roll moistened with distilled water (0.0 mM NaCl–control) and saline water in the other treatments at 250 mM NaCl, obtained by the dissolution of sodium chloride (NaCl), corresponding to 14.61 g L$^{-1}$. We obtained the mM NaCl treatment in preliminary tests. The paper rolls were incubated in a germinator at 25 °C. Germination evaluations, first germination count, and germination percentage were performed four and seven days after sowing, according to Brazilian rules for seed analysis [23].

The lengths of the shoots and roots of normal seedlings were measured at the end of the germination test. Shoot length—SL (measured from the collar to the seedling apex) and primary root length—RL (measured from the collar's base to the root's tip) were measured with a ruler graduated in centimeters.

### 2.4. Dry Mass of Seedlings and Salinity Tolerance index

After the growth measurements, 15 seedlings were placed in kraft paper bags and dried in a forced air circulation oven at 65 °C until they reached a constant weight. Subsequently, they were weighed on a precision scale to obtain shoot (SDM), root (RDM), and total dry mass (TDM), and data were expressed in g plant$^{-1}$. In the calculations of the indices, the total dry mass production of the cultivars was used as the main parameter to determine their tolerance to salt stress.

Total DM data were used to calculate the percentages partitioned between the vegetative organs and the salinity tolerance index by comparing the data of the salt stress treatments with those of the control (EC = 4.13 $\mu$Sm$^{-1}$ at 25 °C), using Equation (1).

$$STI(\%) = \frac{DM \ of \ salt \ stress \ treatment}{DM \ of \ control \ treatment} \times 100, \tag{1}$$

*STI*: salinity tolerance index;
*DM*: dry mass.

### 2.5. Osmotic Homeostasis

Total soluble sugars (TSS), amino acids (AA), and proline (PRO) were obtained from the fresh mass of 10 seedlings. At the time of extraction, the fresh mass was macerated in liquid nitrogen with a crucible and pestle. The plant material sample from each replication was analyzed in triplicate. Then, 0.2 g was weighed, and the material was placed in Eppendorf-type screw cap tubes. Then, 1 mL of 80% ethyl alcohol was added, and the samples were kept in a water bath at 60 °C for 20 min. The material was kept in a centrifuge cooled at 4 °C for 10 min at 10 RPM (procedure performed three times), and the supernatant was collected to quantify the sugars. The total soluble sugars were determined by measuring the absorbance at 620 nm using the anthrone method [25], with glucose as the standard substance, and the results were expressed in mg GLU g$^{-1}$ of fresh mass. The supernatant obtained in the extraction process was used to quantify the amino acid (AA) contents in determining total free amino acids. For this, the acid ninhydrin method was applied, with the absorbance measurement at 570 nm [26], using glycine as the standard substance, and the results were expressed in $\mu$M GLY g$^{-1}$ of fresh mass. Proline determination followed the methodology described by [27]. Proline concentrations were determined based on a standard curve obtained from L-proline and by measuring the absorbance at 520 nm. The results were expressed in $\mu$M PRO g$^{-1}$ of fresh mass.

### 2.6. Statistical Analysis

The data were subjected to analysis of variance (F test); the Scott–Knott test compared the means of discontinuous hydration cycles at a 5% probability level, and Student's *t*-test compared the means of the cultivars at a 5% probability level. Statistical analyses were performed with the computer program SISVAR [28].

## 3. Results

### 3.1. Germination and Seedling Length

The interaction between maize cultivars and pre-germination treatments was significant for the first germination count ($p = 0.0000$), germination ($p = 0.0021$), shoot length ($p = 0.0215$), and root length ($p = 0.0007$) (Table 1).

**Table 1.** F-test and means test (SE, n = 4) for first germination count (FGC), germination (G), shoot length (SL), and root length (RL) for *Zea mays* L. seeds subjected to salt stress tolerance elicitors in three discontinuous hydration cycles (DHCs).

| F-Test (*p*-Value) | | | | |
|---|---|---|---|---|
| **Variation Sources** | **FGC** | **G** | **SL** | **RL** |
| DHCs | 0.0000 | 0.0000 | 0.0000 | 0.0000 |
| Cultivars (C) | 0.0000 | 0.0000 | 0.0145 | 0.0000 |
| DHCs × C | 0.000 | 0.0021 | 0.0215 | 0.0007 |
| **Means-test** | | | | |
| **Cultivars** | **DHCs** | **FGC (%)** | **G (%)** | **SL(cm)** | **RL (cm)** |
| | 1 (control) | 100 ± 0.0 aA | 100 ± 0.0 aA | 7.8 ± 0.38 aA | 16.5 ± 0.12aA |
| | 2 | 47 ± 4.2 dA | 95 ± 1.0 aA | 1.8 ± 0.08 bA | 3.8 ± 0.09 cA |
| | 3 | 77 ± 4.5 cA | 97 ± 1.3 aA | 2.1 ± 0.15 bA | 4.7 ± 0.31 bA |
| BR 206 | 4 | 85 ± 3.3 bA | 97 ± 1.0 aA | 1.9 ± 0.10 bA | 5.2 ± 0.33 bA |
| | 5 | 81 ± 1.9 bA | 96 ± 0.8 aA | 2.1 ± 0.08 bA | 5.0 ± 0.11 bA |
| | 6 | 76 ± 1.5 cA | 98 ± 0.0 aA | 2.2 ± 0.01 bA | 4.9 ± 0.33 bA |
| | 7 | 70 ± 5.5 cA | 98 ± 0.8 aA | 1.9 ± 0.13 bA | 5.0 ± 0.14 bA |
| | 1 (control) | 98 ± 1.3 aA | 99 ± 0.5 aA | 7.9 ± 0.13aA | 14.0 ± 0.51 aB |
| | 2 | 21 ± 3.1 dB | 85 ± 1.7 cB | 1.1 ± 0.13 cB | 3.5 ± 0.30 cA |
| BRS 5037 | 3 | 45 ± 1.3 bB | 92 ± 1.0 bB | 1.7 ± 0.15 bA | 4.5 ± 0.33 bA |
| Cruzeta | 4 | 45 ± 3.3 bB | 92 ± 1.7 bB | 1.3 ± 0.08 cB | 4.9 ± 0.24 bA |
| | 5 | 45 ± 2.9 bB | 92 ± 1.7 bB | 1.8 ± 0.07 bA | 3.7 ± 0.47 cB |
| | 6 | 32 ± 1.4 cB | 86 ± 2.2 cB | 2.2 ± 0.10 bA | 4.8 ± 0.22 bA |
| | 7 | 38 ± 3.8 bB | 92 ± 1.0 bB | 2.1 ± 0.14 bA | 3.8 ± 0.19 cB |

1—0.0 mmol of NaCl (control); 2—250 mmol of NaCl (salt stress); 3—salt stress + DHCs in water; 4—salt stress + DHCs with gibberellic acid (50 μM GA$_3$); 5—salt stress + DHCs with hydrogen peroxide (5 mmol (H$_2$O$_2$); 6—salt stress + DHCs with salicylic acid (50 μM SA); and 7—salt stress + DHCs with ascorbic acid (50 μM ASC). Means followed by the same lowercase letter in the column do not differ by the Scott–Knott test at a 5% probability level, and means followed by the same uppercase letter in the column do not differ from each other by the Student's *t*-test at 5% probability level.

Salt stress reduced the first germination count (FGC) of the BR 206 and BRS 5037 Cruzeta cultivars by 53 and 77 percentage points, compared to the control treatment. For the BR 206 cultivar, three discontinuous hydration cycles (DHCs) improved FGC compared to salt stress. Still, DHCs with GA$_3$ and H$_2$O$_2$ led to results closer to those found in control, with 38 and 34 percentage points more in FGC than salt stress, respectively (Table 1). For the BRS 5037 Cruzeta cultivar, DHCs favored FGC, compared to salt stress. Still, DHCs with elicitors were similar to DHCs with water, except for salicylic acid, which was inferior to DHCs with water (Table 1). The best response to DHCs with elicitors occurred for the BR 206 cultivar, compared to the BRS 5037 Cruzeta cultivar (Table 1).

Salt stress did not affect the germination of the BR 206 cultivar (Table 1). However, salt stress reduced the germination of the BRS 5037 Cruzeta cultivar by 14 percentage points. The DHCs improved germination by an average of seven percentage points compared to salt stress, except for DHCs with salicylic acid, which was similar to salt stress. The best germination levels under salt stress conditions were obtained by the BR 206 cultivar, compared to the BRS 5037 Cruzeta cultivar, in all treatments with salt stress (Table 1).

Seedlings subjected to salt stress had reduced shoot length (SL). For the BR 206 cultivar, treatments with DHCs were not sufficient to overcome the stress caused by salt excess. For the BRS 5037 Cruzeta cultivar, DHCs with elicitors promoted higher SL (mean of 77.3%) than that found for the treatment with salt stress, except for DHCs with gibberellic acid. Only salt stress and DHCs with gibberellic acid promoted different results among cultivars (Table 1).

Root length (RL) was reduced under the salt stress condition. Compared to salt stress, treatments with DHCs favored an average increase of 30.5% in RL for the BR 206 cultivar. For the BRS 5037 Cruzeta cultivar, there was a reduction in RL under salt stress, but the

DHCs favored increments of 28.5% in the root for DHCs with water, 40% for DHCs with $H_2O_2$, and 37.1% for DHCs with salicylic acid (Table 1).

The results presented for SL using DHCs with attenuators for the BR 206 cultivar showed no significant difference caused by salt stress. Still, this cultivar invested significantly in RL under salt stress with attenuators (Figure 2). For the BRS 5037 Cruzeta cultivar, the DHCs with water and DHCs with salicylic acid favored both SL and RL. The DHCs with hydrogen peroxide ($H_2O_2$) and DHCs with ascorbic acid also favored the highest SL for this cultivar. The results indicate that these treatments mitigated salt stress. Even under salt stress, the two cultivars formed seedlings with coleoptile and root development (Figure 2). However, in this study, the formation of coleoptile was reduced in all salt stress treatments compared to the control.

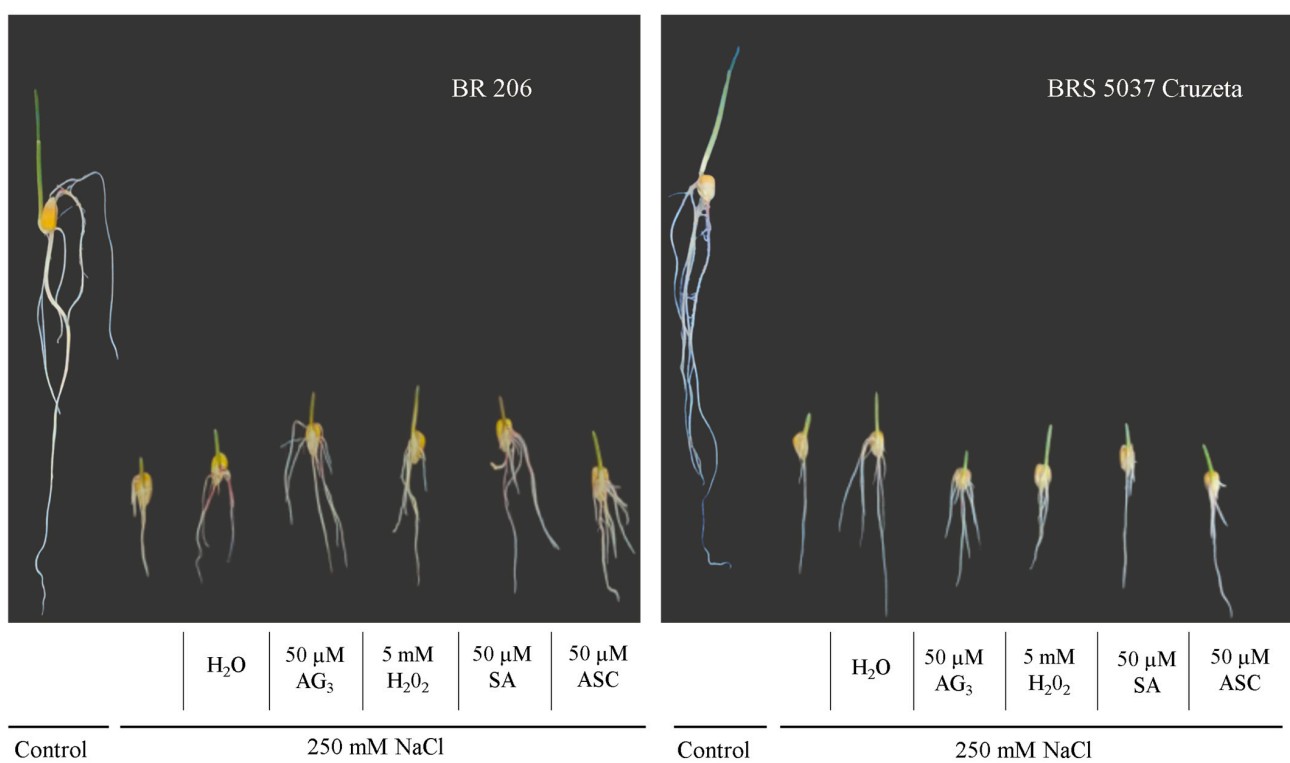

**Figure 2.** *Zea mays* L. seeds subjected to salt stress tolerance elicitors in three discontinuous hydration cycles (DHCs).

### 3.2. Dry Mass of Seedlings and Salinity Tolerance Index

The interaction between cultivars and pre-germination treatments was significant for shoot dry mass ($p = 0.0000$), root dry mass ($p = 0.0000$), total dry mass ($p = 0.0000$), and salinity tolerance index ($p = 0.0000$) (Table 2).

Shoot dry mass (SDM) was reduced in the salt stress, but, for the BR 206 cultivar, the DHCs with water favored a 70.6% increase in SDM compared to salt stress. For the BRS 5037 Cruzeta cultivar, the DHCs with water, DHCs with salicylic acid, and DHCs with ascorbic acid, the SDM increased by 74.5, 83.6, and 87.27%, respectively, compared to salt stress. SDM was higher for the BR 206 cultivar than the BRS 5037 Cruzeta cultivar, regardless of salinity (Table 2).

The BR 206 cultivar had a 68.6% reduction in root dry mass (RDM) under saline stress compared to the control. However, having applied DHCs with gibberellic acid, DHCs with $H_2O_2$, DHCs with salicylic acid, and DHCs with ascorbic acid, the BR 206 cultivar produced 117.6, 104.4, 99.4, and 86.2% more RDM than in the treatment with salt stress, respectively. For the BR 3057 Cruzeta cultivar, salt stress caused a 60.6% reduction in RDM compared to the control, and the treatments with DHCs did not increase this variable. The BR 206 cultivar obtained higher RDM than the BRS 5037 Cruzeta cultivar (Table 2).

The total dry mass (TDM) of the BR 206 cultivar was reduced by 67.1% under salt stress compared to the control. All treatments with DHCs increased TDM in the BR 206 cultivar; however, DHCs with gibberellic acid, DHCs with $H_2O_2$, and DHCs with salicylic acid showed better performance, with values on average 67.5% higher than those of salt stress (Table 2). Compared to the control, the BR 3057 Cruzeta underwent a reduction of 64.4% in TDM under salt stress. In this cultivar, the TDM accumulations in DHCs with water, DHCs with $H_2O_2$, and DHCs with salicylic acid were 45.5, 39.9, and 60.7% higher than those obtained under salt stress, respectively. The BR 206 cultivar produced more TDM than the BR 3057 Cruzeta cultivar, regardless of salinity (Table 2).

Regarding the salinity tolerance index (STI), the cultivars BR 206 and BR 3057 Cruzeta were sensitive (STI < 40%) to salt stress. For the BR 206 cultivar, all treatments with DHCs improved STI, and plants changed from sensitive to moderately sensitive to salinity (40% < STI < 60%) (Table 2). For the BRS 3057 Cruzeta cultivar, all treatments with DHCs, except for DHCs with gibberellic acid, improved STI, and plants changed from sensitive to moderately sensitive to salinity (40% < STI < 60%) (Table 2).

**Table 2.** F-test and means-test (SE, n = 4) for shoot dry mass (SDM), root dry mass (RDM), total dry mass (TDM), and salinity tolerance index (STI) for *Zea mays* L. seeds subjected to salt stress tolerance elicitors in three discontinuous hydration cycles (DHCs).

| F-Test (*p*-Value) | | | | |
|---|---|---|---|---|
| **Variation Sources** | **SDM** | **RDM** | **TDM** | **STI** |
| DHCs | 0.0000 | 0.0000 | 0.0000 | 0.0000 |
| Cultivars (C) | 0.0000 | 0.0000 | 0.0000 | 0.0000 |
| DHCs × C | 0.0000 | 0.0000 | 0.0000 | 0.0000 |
| Means-test | | | | |
| Cultivars | DHCs | SDM mg plant$^{-1}$ | RDM mg plant$^{-1}$ | TDM mg plant$^{-1}$ | STI % |
| | 1 (control) | 35.8 ± 0.5 aA | 50.6 ± 1.9 aA | 86.5 ± 1.8 aA | 100.0 ± 0.0 aA |
| | 2 | 12.6 ± 1.0 dA | 15.9 ± 0.5 dA | 28.5 ± 1.3 dA | 32.9 ± 1.5 dA |
| | 3 | 21.5 ± 0.5 bA | 21.5 ± 1.1 cA | 43.1 ± 0.7 cA | 49.8 ± 0.8 cA |
| BR 206 | 4 | 13.6 ± 0.4 dA | 34.6 ± 1.6 bA | 48.2 ± 1.8 bA | 55.8 ± 2.1 bA |
| | 5 | 15.8 ± 0.8 cA | 32.5 ± 0.8 bA | 48.3 ± 1.3 bA | 55.9 ± 1.5 bA |
| | 6 | 15.0 ± 1.2 cA | 31.7 ± 2.0 bA | 46.7 ± 1.3 bA | 54.0 ± 1.5 bA |
| | 7 | 13.9 ± 1.1 dA | 29.6 ± 1.2 bA | 43.5 ± 1.9 cA | 50.3 ± 2.2 cA |
| | 1 (control) | 25.8 ± 0.6 aB | 25.9 ± 0.7 aB | 51.7 ± 1.1 aB | 100.0 ± 0.0 aA |
| | 2 | 5.5 ± 0.6 cB | 10.2 ± 0.4 bB | 15.8 ± 0.9 cB | 30.5 ± 1.7 eA |
| BRS 5037 Cruzeta | 3 | 9.7 ± 0.4 bB | 13.3 ± 0.8 bB | 23.0 ± 1.0 bB | 44.5 ± 1.8 cB |
| | 4 | 6.5 ± 0.2 cB | 12.5 ± 0.5 bB | 19.1 ± 0.7 cB | 36.8 ± 1.4 dB |
| | 5 | 10.1 ± 0.4 bB | 12.0 ± 0.9 bB | 22.1 ± 1.2 bB | 42.7 ± 2.3 cB |
| | 6 | 11.6 ± 0.3 bB | 13.8 ± 0.7 bB | 25.4 ± 0.8 bB | 49.0 ± 1.6 bB |
| | 7 | 10.3 ± 0.1 bB | 11.6 ± 1.1 bB | 21.9 ± 1.1 bB | 42.4 ± 2.1 cB |

1—0.0 mmol of NaCl (control); 2—250 mmol of NaCl (salt stress); 3—salt stress + DHCs in water; 4—salt stress + DHCs with gibberellic acid (50 µM GA₃); 5—salt stress + DHCs with hydrogen peroxide (5 mmol ($H_2O_2$); 6—salt stress + DHCs with salicylic acid (50 µM SA); and 7—salt stress + DHCs with ascorbic acid (50 µM ASC). Means followed by the same lowercase letter in the column do not differ by the Scott–Knott test at a 5% probability level, and means followed by the same uppercase letter in the column do not differ from each other by the Student's *t*-test at 5% probability level.

### 3.3. Osmotic Homeostasis

The interaction between maize cultivars and pre-germination treatments was significant for the total soluble sugars (*p* = 0.0000), amino acids (*p* = 0.0001), and proline (*p* = 0.0000) (Table 3).

The maize BR 206 cultivar under control, salt stress, and DHCs with salicylic acid treatments obtain the highest total soluble sugars (TSS) levels. However, for the other treatments, the TSS content was lower than that found in control. All treatments with salt stress for the BRS 5037 Cruzeta cultivar led to higher TSS content than the control

(Table 3). In this cultivar, DHCs with salicylic acid favored the highest production of TSS, with 49.6 mg g$^{-1}$ of FM. The BR 206 cultivar produced more TSS than the BRS 5037 Cruzeta cultivar; however, under DHCs with salicylic acid, the contents of these sugars were similar.

**Table 3.** F-test and means-test (SE, n = 4) for total soluble sugars (TSS), amino acids (AA), and proline (PRO) for *Zea mays* L. seeds subjected to salt stress tolerance elicitors in three discontinuous hydration cycles (DHCs).

| F-Test ($p$-Value) | | | |
| --- | --- | --- | --- |
| **Variation Sources** | **TSS** | **AA** | **PRO** |
| DHCs | 0.0000 | 0.0000 | 0.0000 |
| Cultivars (C) | 0.0000 | 0.0000 | 0.0000 |
| DHCs × C | 0.0000 | 0.0001 | 0.0000 |
| Means-test | | | |
| **Cultivars** | **DHCs** | **TSS** mg GLU g$^{-1}$ FM | **AA** μmol GLY g$^{-1}$ FM | **PRO** μmol PRO g$^{-1}$ FM |
| BR 206 | 1 (control) | 50. 1 ± 2.0 aA | 47.5 ± 2.3 dA | 0.7 ± 0.2 eA |
| | 2 | 51.2 ± 0.5 aA | 68.9 ± 4.4 bA | 59.7 ± 0.8 aA |
| | 3 | 47.1 ± 0.4 bA | 65.1 ± 5.1 cA | 48.5 ± 1.7 cA |
| | 4 | 45.1 ± 1.5 bA | 79.9 ± 1.5 aA | 48.3 ± 1.9 cA |
| | 5 | 44.2 ± 2.1 bA | 57.7 ± 1.1 cA | 41.7 ± 2.4 dA |
| | 6 | 49.5 ± 1.4 aA | 78.3 ± 1.9 aA | 58.2 ± 1.5 aA |
| | 7 | 47.5 ± 3.0 bA | 61.3 ± 2.9 cA | 53.2 ± 2.7 bA |
| BRS 5037 Cruzeta | 1 (control) | 25.8 ± 0.9 cB | 22.6 ± 1.1 dB | 0.3 ± 0.01 cA |
| | 2 | 39.9 ± 0.6 bB | 46.9 ± 1.8 cB | 34.8 ± 4.0 bB |
| | 3 | 35.7 ± 1.2 bB | 55.7 ± 1.5 bB | 30.5 ± 0.4 bB |
| | 4 | 39.0 ± 0.7 bB | 68.8 ± 1.5 aB | 45.4 ± 0.9 aA |
| | 5 | 39.7 ± 1.1 bB | 56.5 ± 5.0 bB | 29.8 ± 1.8 bB |
| | 6 | 49.6 ± 1.1 aA | 62.5 ± 5.1 aB | 33.6 ± 0.5 bB |
| | 7 | 37.0 ± 1.0 bB | 67.8 ± 2.5 aA | 32.5 ± 0.7 bB |

1—0.0 mmol of NaCl (control); 2—250 mmol of NaCl (salt stress); 3—salt stress + DHCs in water; 4—salt stress + DHCs with gibberellic acid (50 μM GA$_3$); 5—salt stress + DHCs with hydrogen peroxide (5 mmol (H$_2$O$_2$); 6—salt stress + DHCs with salicylic acid (50 μM SA); and 7—salt stress + DHCs with ascorbic acid (50 μM ASC). Means followed by the same lowercase letter in the column do not differ by the Scott–Knott test at a 5% probability level, and means followed by the same uppercase letter in the column do not differ from each other by the Student's *t*-test at 5% probability level.

Salt stress increased the synthesis of amino acids (AA) by 44.9 and 107.5% in the cultivars BR 206 and BRS 5034 Cruzeta compared to the control, respectively (Table 3). For the BR 206 cultivar, DHCs with gibberellic acid and DHCs with salicylic acid increased the synthesis of AA by 15.9 and 13.7% when compared to salt stress, respectively. For the BRS 5037 Cruzeta cultivar, DHCs with gibberellic acid, DHCs with salicylic acid, and DHCs with ascorbic acid increased the synthesis of AA by 44.6, 33.2, and 44.5% compared to salt stress, respectively (Table 3).

Salt stress increased proline accumulation for the cultivars BR 206 and BRS 5037 Cruzeta by 58.98 and 34.46 μM g$^{-1}$ of FM compared to the control, respectively. For the BR 206 cultivar, the highest proline contents occurred under salt stress and DHCs with salicylic acid. The highest proline accumulation for the BRS 5037 Cruzeta cultivar occurred under DHCs with gibberellic acid. Under salt stress conditions, the BR 206 cultivar produced more proline than the BRS 5037 Cruzeta cultivar, but under DHCs with gibberellic acid, the proline contents were similar between the cultivars (Table 3).

## 4. Discussion

Salt stress causes damage to irrigated agriculture around the world. Research that can improve plant responses to saline stress is essential, but it is challenging to attenuate salinity under severe salinity stress conditions. We found a strategy to do this. We improved maize seedlings' response under severe salt stress (250 mM NaCl) using water memory

and salt stress tolerance elicitors. We found that three discontinuous hydration cycles and different salt stress elicitors mitigated the effect of salt stress and contributed to the improvement of germination, growth, and accumulated dry mass of the maize cultivars. Despite drought tolerance, when exposed to severe saline stress (250 mM NaCl), maize cultivars were sensitive and reduced shoot dry mass by 64–79%, and root dry mass by 61–69% (Figure 3). The use of discontinuous hydration cycles (DHCs) of maize seeds with gibberellic acid (50 μM GA$_3$), hydrogen peroxide (5 mmol (H$_2$O$_2$), and salicylic acid (50 μM SA) promoted losses to shoot dry mass in cultivar BR 206 of 56-62%, and to root dry mass of 32–37% compared to the control. For BRS 5037 Cruzeta, for the DHCs with salicylic acid (50 μM SA), the losses of shoot dry mass were 55%, and 47% of root dry mass compared to the control (Figure 3). Using DHCs in maize seeds under severe stress improved the degree of salinity tolerance of the seedlings, changing them from sensitive to moderately sensitive to salt stress.

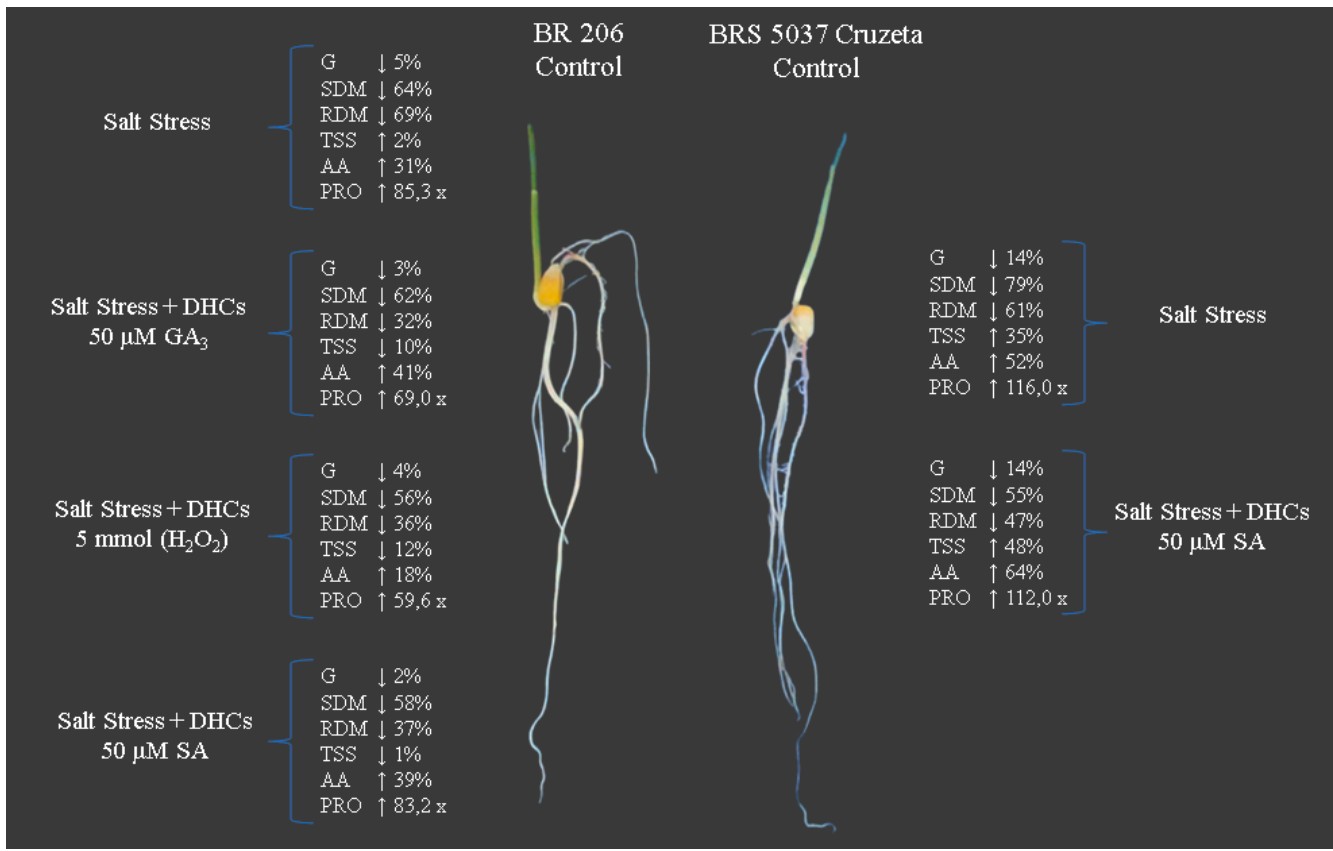

**Figure 3.** Responses of BR 206 and BRS 5037 Cruzeta maize cultivars subjected to salt stress tolerance elicitors in three discontinuous hydration cycles (DHCs) compared to control.

Morphophysiological parameters in both maize cultivars were reduced by salt stress. The results for the BR 206 cultivar were higher than those found for BRS 5037 Cruzeta, except for RL and STI, which were similar. Regarding the FGC of the BR 206 cultivar, DHCs with gibberellic acid stood out, with 85% of seeds germinated. Still, it did not differ statistically from the use of H$_2$O$_2$, which led to 81% of seeds germinating. We have found that this result was observed because gibberellins (GA$_3$) act as stress signaling agents. The responses adopted by plants are for better water absorption, germination, and growth of seedlings, even under unfavorable conditions, such as salt stress [29–31]. Like GA$_3$, H$_2$O$_2$ acts as an oxidative stress signaling agent and, when applied at low concentrations, interacts with hormones that control the germination process [32,33], which explains the higher number of seedlings and uniform germination in FGC in these treatments.

Generally, salt stress conditions reduce the germination percentage of maize seeds [13,29], but this result may vary according to the cultivar. For the BR 206 cultivar, there was no difference in germination among treatments. Even under induced salt stress (250 mM NaCl), all treatments resulted in germination greater than 95%. For the BRS 5037 Cruzeta cultivar, salt stress resulted in a loss of 14 percentage points compared to the control treatment, which proves the variation in germination potential among cultivars. The small changes in the final germination count suggest the need for other parameters to determine the degree of tolerance of maize cultivars. In this context, parameters related to growth and biomass accumulation contribute to explaining maize germination results under salt stress conditions, as they are more sensitive to the effects of salinity [13,34–36].

Maize seedlings respond to salt stress by reducing shoot growth [15,24]. The maize growth decrease occurs due to an osmotic imbalance caused by excess salts. Excess salts decrease water absorption, mobilization of seed nutrient reserves, and elongation of radicle cells, and affect DNA division and synthesis [13,34]. Osmotic stress reduces photosynthetic capacity through stomatal closure and reduction in leaf expansion, consequently decreasing shoot growth [37]. The initial growth is decreased by salt stress; however, as a survival strategy, maize plants invest in root growth before the endosperm reserves are exhausted, corroborating the findings of [24]. Our results reveal that DHCs with tolerance elicitors promote greater shoot length than saline stress, with more significant results in BRS 5037 Cruzeta cultivar, except for DHCs with gibberellic acid. The root length response after DHCs was higher for both cultivars than saline stress, except for DHCs with $H_2O_2$ in BRS 5037 Cruzeta.

Our results reveal that 250 mM NaCl causes toxicity in maize cultivars, causing deficiency in the osmotic adjustment of the plants, even after treatment with DHCs with tolerance elicitors. According to Roy et al. [37], one of the first plant tolerance mechanisms is an osmotic adjustment, regulated by long-distance signals that reduce the length and are triggered before $Na^+$ accumulation, which we verified for maize cultivars. The second strategy occurs via ionic exclusion by decreasing the transport of $Na^+$ and $Cl^-$ in the roots and consequently reducing the accumulation of ions in the leaves. Tolerance to high salt concentrations in leaves is due to the compartmentalization of ions in plant cell vacuoles.

The DHCs with tolerance elicitors contributed to the maize seedlings' tolerance of the salt concentrations, mainly via the investment in the root length from the increase in adventitious roots, primarily in the cultivar BR 206. The increase in the fasciculated root system is significant because seedlings can remove water and nutrients from the environment before the reserves present in the endosperm are exhausted [24]. This response is expressive in the dry mass of seedlings of maize cultivars.

The DHCs with attenuators increased the biomass accumulation of seedlings of maize cultivars subjected to salt stress. The attenuators minimized the effects of salt stress on maize biomass accumulation, but the cultivars showed divergent responses to the attenuators. The DHCs with water favored the increase in SDM in the BR 206 cultivar, but the results of RDM and TDM were lower than those obtained with the other attenuators. On the other hand, the DHCs with gibberellic acid did not increase SDM, RDM, and TDM for the BRS 5037 Cruzeta cultivar. The highest biomass accumulations occurred with DHCs with gibberellic acid, hydrogen peroxide, and salicylic acid for the BR 206 cultivar, and DHCs with water, DHCs with hydrogen peroxide, DHCs with salicylic acid, and DHCs with ascorbic acid for the BRS 5037 cultivar.

Recent studies with discontinuous hydration cycles in water have induced the tolerance of forest species to water stress at the morphophysiological level [17,38–40]. Discontinuous hydration cycles allow the seeds to improve the expression of vigor in an adverse environment, so their hydration and dehydration favor stress acclimation. The authors report that, during this process, metabolic, energetic, and respiratory activities are initiated by the mobilization of reserves, along with β-oxidation of fatty acids, nitrogen mobilization, and the improvement of membrane permeability. These improvements favor embryonic growth and root emergence in plants under water stress, which is new to salt stress.

Discontinuous hydration cycles improved the tolerance of the cultivars BR 206 and BRS 5037 Cruzeta to osmotic restriction induced by excess salts. Maize seeds showed high sensitivity to salt stresses of 250 mM NaCl, with biomass losses close to 70% compared to the control. However, in the DHCs with attenuators, the seedlings changed from sensitive to moderately sensitive to salinity, with biomass losses lower than 60% for BRS 5037 Cruzeta and lower than 50% for BR 206. The increase in dry mass occurred after DHCs with salt stress tolerance elicitors, which corroborates the research results on using DHCs with water to mitigate stress [17–19]. The response demonstrates that the dehydration process cannot erase this hydration memory, which is already a cause of stress, and the seeds have greater tolerance to the new stress [41]. The dehydration process naturally damages the lipid membranes. The diacylglycerol present in plant cells (plastid and endoplasmic reticulum) is a precursor in synthesizing glycerolipids that disorganizedly cause electrolyte leakage [40] that can affect seedling development. Seeds invest their metabolic energy in membrane repair rather than growth to reduce this damage.

Salt stress naturally increased the synthesis of amino acids compared to the control; however, the DHCs with gibberellic acid and DHCs with salicylic acid further increased the synthesis of amino acids, mainly proline, in both cultivars. The BR 206 cultivar obtained higher amino acid and proline levels than the BRS 5037 Cruzeta cultivar. This greater capacity for the synthesis of osmolytes of the BR 206 cultivar is related to its higher tolerance. Osmotic adjustment in plant cells ensures the maintenance of water entry and cell turgor, limiting the damage caused at the beginning of stress [42,43].

An important fact is that the synthesis of sugars was not altered between control and salt stress in both cultivars, indicating that the osmotic adjustment of maize occurs mainly via the increase in amino acid synthesis. However, under DHCs with water, $H_2O_2$, and ascorbic acid, there was a decrease in sugar synthesis to the detriment of increased amino acid synthesis. Only under DHCs with gibberellic acid and salicylic acid did the increase in the synthesis of amino acids not coincide with a decrease in the synthesis or degradation of sugars, which are more efficient in inducing osmotic adjustment in maize.

DHCs with tolerance elicitors potentiated the responses adopted by maize cultivars under saline stress. Germination and growth of BRS 5037 Cruzeta maize decreased more under saline stress than BR 206 maize. Therefore, this cultivar is more sensitive to saline stress. Cultivar BR 206 responded better to salt stress tolerance elicitors, such as gibberellic acid, salicylic acid, and hydrogen peroxide. However, cultivar BRS 5037 Cruzeta responded only to treatment with salicylic acid because it is more salinity-sensitive than BR 206 maize (Figure 3). Salicylic acid is a salt stress tolerance elicitor indicated for salinity-sensitive maize cultivars. Khan et al. [44] indicate that salicylic acid is a phenolic compound that favors the growth and development of plants through regulation and production. In cultivar BR 206, elicitors promoted osmotic adjustment, verifying significant increases in AA and PRO in DHCs with GA3, $H_2O_2$, and SA compared to control and saline stress without elicitors. However, in cultivar BRS 5037 Cruzeta, DHCs-SA significantly increased TSS, AA, and PRO compared to control and salt stress without elicitors. The DHCs with salicylic acid allowed the shoot and root length results and the accumulation of amino acids to be similar between the two cultivars. However, the TSS accumulation in the salinity-sensitive cultivar treated with DHCs-SA was significant for acclimatization to salt stress. Our results confirm that salicylic acid favors plant growth and development [44]. The salicylic acid promotes an increase in proline production, increasing osmotic adjustment, allowing for more water absorption and triggering antioxidant enzyme activity [30,45]. This result may be more apparent when the species is susceptible to saline stress, as observed for BRS 5037 Cruzeta cultivar.

## 5. Conclusions

Salt stress (250 mM NaCl) reduced maize germination, growth, and dry mass accumulation, and the BR 206 cultivar was more salinity-tolerant than the BRS 5037 Cruzeta cultivar. Discontinuous hydration cycles with water failed to increase the salt stress tolerance of

maize seeds. However, discontinuous hydration cycles with gibberellic acid, hydrogen peroxide, and salicylic acid promoted salt stress tolerance in maize. Discontinuous hydration cycles with saline stress tolerance elicitors improve germination, growth, and dry mass accumulation of maize under saline stress, mainly by inducing the synthesis of the osmoprotectant, such as proline, amino acids, and sugars. The osmotic adjustment in the salinity-tolerant cultivar—BR 206—occurred via the increase in amino acids and proline, and in the salinity-sensitive cultivar—BRS 5037 Cruzeta—via the increase in sugars, amino acids, and proline. The salinity-tolerant cultivar—BR 206—responded to three stress elicitors: gibberellic acid, hydrogen peroxide, and salicylic acid. However, the salinity-sensitive cultivar—BRS 5037 Cruzeta—responded only to salicylic acid. Salicylic acid is the most suitable elicitor for discontinuous hydration cycles in maize seeds aiming to increase salt stress tolerance.

**Author Contributions:** K.T.O.P., S.B.T., E.P.d.P. and F.V.d.S.S. contributed to the study conception and design. Conduction of the experiment, data collection, and analysis were performed by K.T.O.P., T.R.C.A. and M.L.d.S.N. K.T.O.P. and F.V.d.S.S. wrote the first draft of the manuscript. S.B.T., E.P.d.P., M.L.d.S.N., J.B.V., L.S.S., C.P.B., T.D.C.P., M.F.N., N.d.s.D. and F.V.d.S.S. contributed to the manuscript revision, read, and approved the final version. All authors have read and agreed to the published version of the manuscript.

**Funding:** This research was funded by Coordenação de Aperfeiçoamento de Pessoal de Nível Superior, grant number 001.

**Data Availability Statement:** All data are presented in the paper.

**Conflicts of Interest:** The authors declare no conflict of interest.

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
