# Peer review of "Discontinuous Hydration Cycles with Elicitors Improve Germination, Growth, Osmoprotectant, and Salt Stress Tolerance in Zea mays L."

_agriculture, doi:10.3390/agriculture13050964_

Round 1

Reviewer 1 Report

The text is interesting and valuable. There are number of points that you may pay attention to improve the text. These are as follows:

A-Line 18 'your association'  ......please correct

B-Line 112, please give Zea mays as in italics,

C-Line 119, GA3 should be written appropriately...

D-Line 133 and further, ın the results you gave the shoot lenght as SL or total soluble sugar as TSS. You should give these abbreviations in Materials and Methods.  Then readers will be aware of the terms later on.

E-Figure 2, It will be better saying 4th day count rather than first count. 7th day germination than just germination.

F-Please check all figures GA3 was written as AG3 this is big mistake ..... Be careful.....

G-Lines 209 and 212, the text is lower case than the other part of the text.

H-Figure 5 please give Zea mays is in italic,

I-Line 397, ıt can not be biomass increase in this article since seeds did not become a develeoped plant yet. So biomass can not be produced.....

J- Why 2 hours hydration and why three cycles ? Is there a preliminary finding that is based on ? Can they be higher number of cycles or lower be more effective . Any idea ? May be add in discussion.

The text can be published after minor revision.

Author Response

Dear Reviewer 1,

We appreciate your suggestions and take your suggestions into account.

We have adjusted the summary on lines 18-19.

We added variable abbreviations in material and methods in lines: 142-174.

We have adjusted the AG3 term to GA3.

We have corrected the font size throughout the manuscript.

We have set Zea mays to be italicized in figure 5.

We changed the term biomass to dry mass.

The description of hydration cycles has been improved in lines 128-132.

All changes you and other reviewers suggested are highlighted in red in the text.

We appreciate it if you recommend our article for publication.

Cordially,

Francisco Vanies da Silva Sá, D.Sc. & Professor

Reviewer 2 Report

In this manuscript, the authors reported that discontinuous hydration cycles improved salt tolerance of maize seedlings. The results are of help for understanding the effect of DHCs on stress response of crops. Nevertheless, there are still some points which need to be addressed:

 Major points

1.     Although the authors have provided some novel data, the overall novelty of the manuscript is still not satisfying.

2.     For the two maize cultivars, the author mentioned that BR206 is drought tolerant, and BRS5037 is adapted to semi-arid regions. The information is not sufficient to explain why they choose the two cultivars. Since the important differences (e.g. genetic background, growth and development behavior, response to various stresses) between the two cultivars were not mentioned, it is difficult to understand why the authors choose them and how can they conclude based on comparing the difference of the two cultivars’ seedlings.

 Minor points

1.     Line 21-25, it is unnecessary to show so detailed experiment design information in the abstract.

2.     In the introduction, research progress about plant stress tolerance were not adequately reviewed. The authors only mentioned some simple, well-known knowledges.

3.     Line 95, “The BRS 5037…is very early,…”, it is not understandable, early for what?

4.     In Section 3.1, 3.2, and 3.3, the first paragraph is very hard to understand. “The interaction between maize cultivars and pre-germination treatments was significant”, can the two cultivars (or various treatments) interact?

5.     In Figure 2, 3, 4, and 5, GA3 was mistakenly spelled as “AG3”, H2O2 was mistakenly spelled as “H202”.

6.     Line 230-238, is TDM the sum of SDM and RDM? If so, it is unnecessary to show the data in figure 3 and describe these results in the text.

Author Response

Dear Reviewer 2,

We appreciate your suggestions and accept any possible requests.

We have improved the description of cultivars in lines 96-102.

We adjusted the AG3 term to GA3 and standardized the H2O2.

We used a two-way ANOVA to explain the cultivar and treatment factors. And we got interaction between Factor 1 x Factor 2.

We chose to keep the description of the treatments in the Summary and the total dry matter variable.

All changes you and other reviewers suggested are highlighted in red in the text.

We appreciate it if you recommend our article for publication.

Cordially,

Francisco Vanies da Silva Sá, D.Sc. & Professor

Round 2

Reviewer 2 Report

The authors revised the manuscript by adding some information about the two maize cultivars and correcting some spelling mistakes. The manuscript looks a little better than its original version. Nevertheless, the novelty of the results is still not satisfying. 

Author Response

Dear Reviewer,

We have adjusted the text as per Editor’s suggestions. The statistics were performed as described in the material and methods, but we only presented graphs with the average tests in the previous version. In the current version, we present data in tables with ANOVA, standard error, Scott-Knott test, and Student's t-test.

We adjusted the text for clarity on lines: 17-18, 34, 37, 43-43, 45, 54, 61, 81, 91-92, 99, 111, 113, 127-130, 141-142, 149, 152, 164-167, 177, 190-201, 224, 229-237, 244-255, 261, 264, 269, 274, 278, 282, 284, 288-297, 299, 304, 311, 319-326, 359, 365, 408, and 414-415.

We underline all changes in red in the text.

We appreciate you recommending our article for publication.

Best regards,

Francisco Vanies da Silva Sá, D.Sc. and Professor.
